# Different Radiation Tolerances of Ultrafine-Grained Zirconia–Magnesia Composite Ceramics with Different Grain Sizes

**DOI:** 10.3390/ma12172649

**Published:** 2019-08-21

**Authors:** Wenjing Qin, Mengqing Hong, Yongqiang Wang, Jun Tang, Guangxu Cai, Ran Yin, Xuefeng Ruan, Bing Yang, Changzhong Jiang, Feng Ren

**Affiliations:** 1School of Physics and Technology, Center for Ion Beam Application, Hubei Nuclear Solid Physics Key Laboratory and MOE Key Laboratory of Artificial Micro- and Nano-Structures, Wuhan University, Wuhan 430072, China; 2School of Physics and Electronics, Key Laboratory of Low Dimensional Quantum Structures and Quantum Control, Hunan Normal University, Changsha 410081, China; 3The Institute of Technological Sciences, Wuhan University, Wuhan 430072, China; 4Materials Science and Technology Division, Los Alamos National Laboratory, Los Alamos, NM 87545, USA; 5School of Power and Mechanical Engineering, Wuhan University, Wuhan 430072, China

**Keywords:** ZrO_2_–MgO, ultrafine grain, irradiation, He bubble, phase transformation

## Abstract

Developing high-radiation-tolerant inert matrix fuel (IMF) with a long lifetime is important for advanced fission nuclear systems. In this work, we combined zirconia (ZrO_2_) with magnesia (MgO) to form ultrafine-grained ZrO_2_–MgO composite ceramics. On the one hand, the formation of phase interfaces can stabilize the structure of ZrO_2_ as well as inhibiting excessive coarsening of grains. On the other hand, the grain refinement of the composite ceramics can increase the defect sinks. Two kinds of composite ceramics with different grain sizes were prepared by spark plasma sintering (SPS), and their radiation damage behaviors were evaluated by helium (He) and xenon (Xe) ion irradiation. It was found that these dual-phase composite ceramics had better radiation tolerance than the pure yttria-stabilized ZrO_2_ (YSZ) and MgO. Regarding He^+^ ion irradiation with low displacement damage, the ZrO_2_–MgO composite ceramic with smaller grain size had a better ability to manage He bubbles than the composite ceramic with larger grain size. However, the ZrO_2_–MgO composite ceramic with a larger grain size could withstand higher displacement damage in the phase transformation under heavy ion irradiation. Therefore, the balance in managing He bubbles and phase stability should be considered in choosing suitable grain sizes.

## 1. Introduction

Ceramics with high radiation tolerance are often used for radioactive waste management as inert matrix fuel (IMF) and cladding materials for fission reactors [1]. Zirconia (ZrO_2_) is considered to be a promising material for IMF to burn minor actinides, plutonium and to immobilize high-level nuclear waste because of its excellent physical and chemical stability, including thermal and radiation stability and low neutron capture cross-section [2]. To enhance the radiation tolerance of ZrO_2_ under various energetic particles’ (He, Xe, Cs, and so on) irradiation environments, one of the effective methods is to reduce its grain size. Nanostructured materials contain abundant grain boundaries (GBs) or phase interfaces, which can effectively act as “sinks” to attract, absorb, and annihilate defects [3,4]. Refining the grain size to increase the density of GBs is a widely used method to improve the radiation tolerance performance of materials. Compared to bulk nickel (Ni), nanocrystalline Ni with an average grain size of 55 nm can significantly reduce the density and size of defect clusters induced by irradiation [5]. Nanocrystalline MgGa_2_O_4_ with an ~8 nm size can tolerate 96 displacements per atom (dpa) without any signs of amorphizing, while the large grain (~10 μm) counterpart begins to amorphize just at 12 dpa [6]. The intrinsic reason for the radiation tolerance of GB-reinforced materials is that the GBs have a “loading–unloading” effect, making the radiation-induced damage “self-healing”. A low energy barrier or even barrier-free region is formed around interstitial atoms when the interstitial atoms are loaded in or near the GBs, which promotes the annihilation of vacancy [7,8]. 

However, it was also found that a small grain size did not always indicate high radiation tolerance. Lu et al. found that monoclinic ZrO_2_ with a grain size less than 40–50 nm was much easier to amorphize than the bulk component under swift heavy ion irradiation [9]. Radiation tolerance performance of the nanostructured material is size-dependent, according to the theory of kinetics and thermodynamics [10]. The ideal grain size for nanocrystals with “self-healing” behavior may be a delicate balance between the excess free energy from GB, the free energy of phase transition and defect formation, and the healing kinetics [11]. Since there are no available data for the ideal grain size of nanostructured ZrO_2_, it is worthy of further exploration for the purpose of avoiding these problems. 

Furthermore, most nanocrystalline grains will undergo rapid coarsening under high temperature or ion irradiation, decreasing the density of GBs and effectively weakening the sink of GBs [12,13]. Because multiphase composition can increase the coupling diffusion distance of the same component grain, it can reduce the driving forces of grain growth and improve the stability of the GBs [14]. Magnesia (MgO), with a low neutron capture cross-section, high thermal conductivity, and high electrical conductivity, is regarded as a promising contender in the nuclear energy field [15]. In this work, to improve the stability of ZrO_2_, we added MgO to the ZrO_2_ and prepared ultrafine-grained ZrO_2_–MgO dual-phase composite ceramics with two different grain sizes by spark plasma sintering (SPS). The evolution behavior of defects and the mechanisms of radiation tolerance in these dual-phase composite ceramics were investigated by irradiation of light helium (He) ion and heavy xenon (Xe) ion, respectively. 

## 2. Materials and Methods 

### 2.1. Material Preparation

ZrO_2_–MgO composite ceramics were prepared by mixing monoclinic ZrO_2_ powders (99.99%) and cubic MgO powders (99.99%) with a volume ratio of 1:1, and sintered by using SPS. The average grain sizes of the ZrO_2_ and MgO powders were ~50 nm, and they were purchased from Aladdin Reagents Co. Ltd (Shanghai, China); the relevant grain orientations can be seen in Appendix A. To obtain different grain sizes and control the excessive growth of the grains, the mixing ZrO_2_–MgO powders were sintered at 1373 K and 1623 K with the same pressure (80 MPa), respectively. In addition, the individual pure ZrO_2_ and MgO ceramics were used as comparison samples. Large-grained MgO ceramics with dozens of microns were sintered at 1673 K with 100 MPa pressure. The cubic {110}-oriented, single-crystal, yttria-stabilized ZrO_2_ (YSZ) was supplied by Kejing Materials Company in Hefei, China. All the sintered samples were annealed in vacuum at 1073 K for 4 h to remove infiltrated carbon.

### 2.2. Light He^+^ Ion Irradiation

To explore the effect of grain boundary on the He atom behavior, the ultrafine-grained ZrO_2_–MgO composites with different grain sizes, YSZ, and MgO were irradiated at 773 K by 50 keV He^+^ ions to fluences of 3 × 10^16^, 5 × 10^16^, and 1 × 10^17^ ions/cm^2^, respectively. The ion flux was controlled at about 1.8 × 10^13^ ions/cm^2^s. The He^+^ ion irradiation experiments were performed using the 200 keV ion implanter (LC22-100-01, Beijing Zhongkexin Electronics Equipment Co., Ltd, Beijing, China) at the Center for Ion Beam Application, Wuhan University. The displacement per atom (dpa) and He concentration of the samples irradiated to a fluence of 1 × 10^16^ ions/cm^2^ were calculated by SRIM-2013 with a “Quick Kinchin-Pease” mode [16], where the threshold displacement energies (E_d_) of Zr, Mg, and O atoms were all set to 40 eV, and the theoretical density values of ZrO_2_–MgO, YSZ, and MgO (4.63, 6.10, and 3.58 g/cm^3^) were used, respectively [17,18,19]. The calculation results of the ZrO_2_–MgO composite are shown in Figure 1a. The maximum dpa was about 0.21 at the depth of around 228 nm, and the peak He concentration was about 0.72 at. % at the depth of around 282 nm. The relevant calculation results of the other two comparison samples are shown in Appendix A.

### 2.3. Heavy Xe^23+^ Ion Irradiation

To further investigate the influence of cascade damage on samples, 6.4 MeV Xe^23+^ ions were used to irradiate the different-grain-sized ZrO_2_–MgO composites, YSZ, and MgO at room temperature (RT) on the 320 kV platform for multi-disciplinary research with highly charged ions at the Institute of Modern Physics, Chinese Academy of Sciences (CAS). The fluences were 5 × 10^15^, 1 × 10^16^, and 3 × 10^16^ ions/cm^2^, respectively. The results of SRIM calculation are shown in Figure 1b, which were also simulated by a “Quick Kinchin-Pease” mode. For the ZrO_2_–MgO composite, the maximum displacement per atom reached up to 15 dpa at the depth of around 1.27 µm when the fluence was 1 × 10^16^ ions/cm^2^. Both electronic energy loss and nuclear energy loss contributed to the irradiation damage, because they were in the same order of magnitude [20].

### 2.4. Characterization Method

The changes of microstructures were observed by transmission electron microscopy (TEM, JEOL JEM-2010 (HT), JEOL, Ltd., Tokyo, Japan). Cross-sectional TEM samples were prepared using a conventional precision ion polishing system (Gatan 691, Gatan, Inc., Pleasanton, CA, USA). Raman scattering spectra were characterized by a commercial Raman microscope (HR800, Horiba, Ltd., Longjumeau, France) using an Ar^+^ ion laser excitation source (λ = 488 nm). Crystalline structures of the pristine and irradiated samples were identified by grazing incidence X-ray diffraction (GIXRD, Rigaku, Japan) on a SmartLab 9 kW with Cu *Kα* radiation. The X-ray penetration depth in the ZrO_2_–MgO composites versus grazing incidence angle (α) were estimated geometrically, critical angle (α_c_), and total external reflection theory [21,22], as shown in Appendix A. Considering the damage depth induced by 50 keV He^+^ ions, 0.5° grazing incidence angle was chosen.

## 3. Results and Discussion

### 3.1. Properties of Pristine Samples

Figure 2a,b show the bright-field TEM images of the ZrO_2_–MgO composite ceramics sintered at 1373 K and 1673 K, respectively. The white grain is MgO and the black is ZrO_2_, judged by selected area electron diffraction (SAED). The average grain size of the ZrO_2_–MgO composite sintered at 1373 K was 110.2 ± 45.1 nm, while that of the ZrO_2_–MgO composite sintered at 1673 K was 349.0 ± 76.2 nm, as shown in Figure 2c. For convenience of marking, they are labeled as SGs and LGs according to the grain size, respectively. Figure 2d shows the corresponding crystalline structures. Clearly, the ZrO_2_ in the SGs and LGs was a mixture of monoclinic and tetragonal phase, but the proportion of the tetragonal phase was different. The Gravice and Nicholson method was used to estimate the proportion of tetragonal phase in the two mixed phases of ZrO_2_ [23]. The relevant expression is as follows:(1)$$C_{t}=\frac{I_{t}(101)}{I_{m}(111)+I_{t}(101)+I_{m}(-111)}$$ where *I* refers to the intensity of diffraction peak obtained from the GIXRD. The proportions of tetragonal phase in the SGs and LGs were 15.1% and 24.7%, respectively. Therefore, there were two different aspects between the SGs and LGs: (i) the average grain size; (ii) the proportion of the tetragonal phase.

### 3.2. Microstructure Evolution under He^+^ Ion Irradiation 

Figure 3 shows the cross-sectional TEM images of SGs, LGs, MgO, and YSZ irradiated at 773 K by 50 keV He^+^ ions to fluences of 3 × 10^16^, 5 × 10^16^, and 1 × 10^17^ ions/cm^2^, respectively. These images were obtained from the regions of peak damage depth. At the lowest fluence, it is obvious that there were many large cracks in the YSZ (Figure 3d). Large numbers of He bubble chains were observed in the MgO (Figure 3c), and large, ribbon-like bubbles appeared in the LGs (Figure 3b), while only very small bubbles were found in the SGs (Figure 3a). Note that the ribbon-like He bubbles in the LGs only existed in the grains of ZrO_2_. At the same irradiation condition, the distribution of He bubbles in the MgO and ZrO_2_ (or YSZ) was obviously different; the distribution range with obvious He bubbles in the MgO was larger than that in the YSZ. This was because the migration barrier of the interstitial atoms in the MgO was much lower than that in the ZrO_2_, which resulted in the diffusion of the interstitial atoms in the MgO to a deeper position, thereby also leading to a wide distribution of residual vacancies [24]. In addition, vacancies in the MgO were immobile at this current irradiation temperature (773 K), and they begin to mobilize only when the temperature was higher than 873 K [19]. Thus, at the low fluence, it is possible that the implanted He atoms were trapped by these isolated vacancies with wide distribution and stabilized the vacancies formed there, which further reduced the accumulation of He bubbles [25]. This led to the absence of the large, ribbon-like He bubbles in the grains of MgO. With the increase of fluence to 5 × 10^16^ ions/cm^2^, cracks were also formed in the LGs and MgO. However, no cracks were found in the SGs, although relatively large-sized He bubbles were observed. When the fluence was further increased to 1 × 10^17^ ions/cm^2^, small cracks appeared in the SGs, and the number and size of the cracks become greater in the LGs. Compared with the pure MgO and YSZ, the dual-phase composite ceramic had better radiation tolerance in managing He bubbles. Meanwhile, for the dual-phase composite ceramic, large He bubbles were more likely to form in the ZrO_2_ grains, and the SGs with relatively smaller grain size had better management of He bubbles than the LGs. 

To further understand the lattice changes induced by He^+^ ion irradiation, the GIXRD patterns of the pristine SGs (Figure 4a) and LGs (Figure 4b) irradiated to fluences of 3 × 10^16^, 5 × 10^16^, and 1 × 10^17^ ions/cm^2^ were measured and are shown in Figure 4. To facilitate observation, the diffraction patterns (angles (2*θ*) between 26° and 35°, which are the main peak positions of monoclinic and tetragonal phases) have been magnified. Clearly, no amorphization and phase transformation were found in the SGs and LGs after He^+^ ion irradiation. Thus, the specific orientation corresponding to the diffraction peak can be referred to Figure 2d. With increase of the fluence, the intensities of the main diffraction peaks were gradually reduced and the peaks became broader. The deviations of all the diffraction peaks were very different with the increasing fluence. 

Because there were large numbers of diffraction peaks in the ZrO_2_–MgO composite ceramic, to facilitate the analysis, three main representative peaks, {111}*_m_*, {101}*_t_*, and {200}*_MgO_*, were selected for comparison to study the influence of irradiation. The diffraction angle values of these three main peaks that varied with increase of the fluence are listed in Table 1, and the diffraction angle was measured by the full width at half maximum (FWHM) of the peak. The diffraction peaks {111}*_m_* of the irradiated SGs and LGs both shifted to a lower angle and the shifts were small relative to their pristine samples. The offset of LGs increased gradually with the increase of fluence, while the offset of SGs increased gradually and then decreased at a fluence of 1 × 10^17^ ions/cm^2^. The variation tendency of the diffraction peaks {101}*_t_* and {200}*_MgO_* were similar to that of peak {111}*_m_* in the SGs. However, for the irradiated LGs, the variation of the peak {101}*_t_* and {200}*_MgO_* was very different from that of the peak {111}*_m_*. The peak {101}*_t_* shifted to the right after irradiation and did not change with the increase of fluence, while the peak {200}*_MgO_* shifted first to the right and then to the left as the fluence increased. From the above analysis, it can be seen that the shift tendency of each peak under different damage conditions was different, which means that its ability to resist He^+^ ion irradiation was different. Referring to Figure 3, it can be considered that the left-shift of the diffraction peak was produced due to the lattice swelling induced by He bubbles, while the subsequent right-shift of the peaks in SGs was mainly due to the stress relaxation induced by crack formation, as shown in the TEM image [26,27]. For the LGs, although the cracks formed under high He concentration, this did not affect the continued swelling of the monoclinic grains (peak {111}*_m_*), which may have been due to the large grain size preventing the release of He in time. Stress relaxation occurred in the peaks {101}*_t_* and {200}*_MgO_*, possibly because of the combination of the crack formation and thermal effects (long-time high temperature irradiation).

### 3.3. Heavy Xe^23+^ Ion Irradiation

Raman scattering is a powerful technique used to characterize the microstructural evolution after irradiation. Figure 5 shows the Raman spectra of the pristine SGs and LGs irradiated by 6.4 MeV Xe^23+^ ions to the fluences of 5 × 10^15^, 1 × 10^16^, and 3 × 10^16^ ions/cm^2^, respectively. Group theory predicted 18 Raman active modes (9A_g_ + 9B_g_) for the monoclinic ZrO_2_, 6 Raman active modes (A_1g_ + 2B_1g_ + 3E_g_) for the tetragonal ZrO_2_, and only 1 Raman active mode for the cubic ZrO_2_, which appeared at around 600 cm^−1^ [28]. By comparing the Raman spectra of different phases of ZrO_2_ [29], it was found that the pristine SGs and LGs were mainly composed of the monoclinic phase, as well as the tetragonal phase of ZrO_2_. No Raman band of MgO was observed in this wavenumber range. Among the Raman bands of the pristine samples, only three bands represented the tetragonal phase, which have been marked in Figure 5, while the others corresponded to the monoclinic phase. Compared with the pristine SGs, the bands in the pristine LGs were shifted to lower wavenumbers, and the spectral peaks of some bands become less obvious, such as at around 186, 310, 346, 503, 539, and 621 cm^−1^. Unexpectedly, huge spectral changes occurred in the irradiated SGs and LGs. The peak intensities of spectral bands sharply declined and decreased with the increase of the irradiation fluence. At the lowest fluence of 5 × 10^15^ ions/cm^2^ (corresponding to peak damage of 7.5 dpa), a new band at 710 cm^−1^ appeared in the LGs, and the other bands corresponding to the monoclinic and tetragonal phases still existed. However, in the SGs, many bands, such as at 148, 181, 224, 310, 336, 351, 506, 541,566, and 645 cm^−1^, all disappeared. The bands at 188 and 619 cm^−1^ were formed by the left-shift of the corresponding original bands. Meanwhile, two new bands appeared at 259 and 712 cm^−1^. The intensities of bands at 385 and 478 cm^−1^ became very low. These band changes that occurred in the irradiated SGs to 7.5 dpa were also observed in the LGs when the fluence increased to 1 × 10^16^ ions/cm^2^ (15 dpa). With the fluence continuously increasing to 3 × 10^16^ ions/cm^2^ (45 dpa), the bands in the SGs and LGs were only slightly shifted to the left, without significant changes. It should be noted that no amorphization was found in the irradiated SGs and LGs under such high fluence.

For the new bands at 710 and 712 cm^−1^, they did not represent any vibrational mode of the known ZrO_2_ phases, but rather revealed the vibrational mode induced by defects in the ZrO_2_ phases. In addition, there have been two different suggestions made about the bands at between 250 and 261 cm^−1^. Some researchers have suggested that these vibrational modes were caused by defects [29], while others have suggested that they represent the vibrational modes of the tetragonal phase [28,30,31]. Combined with the Raman spectra of the pristine samples, we prefer the second argument, that these bands are obtained from the left offset of the 267 and 265 cm^−1^ bands with the increase of the fluence, respectively. By comparing the four main Raman bands after irradiation, it can be seen that the bands gradually shifted towards the left with increasing fluence, due to the increase in the residual stress and the generation of defects induced by irradiation [28,32]. In addition, the deviation of bands corresponding to tetragonal phase was greater than that of the monoclinic phase, which means that the tetragonal phase was more sensitive to the change of the stress field induced by irradiation than the monoclinic phase [32]. 

As a whole, with the increase of irradiation damage, most of the monoclinic active modes in the SGs and LGs disappeared, while the relative intensities corresponding to the active modes of tetragonal phase gradually increased, which means that the partial monoclinic phase in the SGs and LGs was transformed into a tetragonal phase. By analyzing the bonds (Zr–Zr, Zr–O, or O–O) corresponding to each active mode, it was found that the disappeared monoclinic active modes were mainly composed of O–O bonds, which indicates that the O–O bonds were damaged the most seriously during the Xe^23+^ ion irradiation [33]. This is consistent with what was described by Simeone et al., that the partial phase transformation was due to the nanometric scale strain field caused by defects in the oxygen sublattice. The strain field can reduce the phase transition temperature and quench the high temperature tetragonal phase [28,34]. Notably, the LGs could withstand higher displacement damage than the SGs in the phase transformation. It is proposed that the presence of low grain size or a high-density grain boundary, and their inherent sub-stoichiometry, may enhance the disorder effect under heavy ion irradiation [35]. The above results show that the ultrafine-grained ZrO_2_–MgO composite ceramic with a large grain size had better radiation tolerance under high energy heavy ion bombardment.

Therefore, it was difficult to directly find out which grain size of composite ceramics had better radiation tolerance performance under ion irradiation with different energy by comparing the results of the above He^+^ and Xe^23+^ ion irradiation experiments.

## 4. Conclusions

In this work, dual-phase ZrO_2_–MgO composite ceramics with refined grain sizes were prepared by SPS to improve their radiation tolerance performance as IMF. Under He^+^ ion irradiation, the ZrO_2_–MgO composite ceramics had a higher ability to manage He bubbles, relative to the pure MgO and YSZ. In the composite ceramic, the ribbon-like He bubbles were more likely to be formed in the ZrO_2_ grains. Meanwhile, the ZrO_2_–MgO composite ceramics with smaller grain sizes had better ability to manage He bubbles than those with large grain sizes. However, under heavy Xe^23+^ ion irradiation to high displacement damage, the ZrO_2_–MgO composite ceramic with larger grain sizes could withstand higher displacement damage than small grain sizes in the phase transformation. Therefore, it was feasible to enhance the radiation tolerance of the ceramics by grain refinement under low irradiation displacement damage, but not at high displacement damage. For the actual application of ZrO_2_–MgO composite ceramics, the balance between managing He bubbles and phase stability should be considered by choosing a suitable grain size.

## Figures and Tables

**Figure 1 materials-12-02649-f001:**
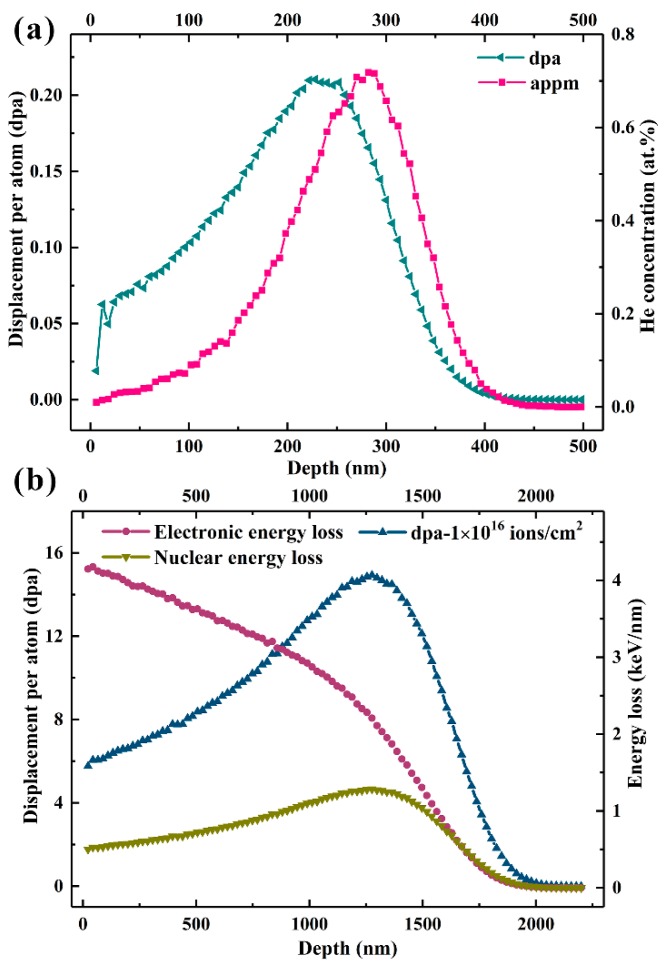
The SRIM-2013 results of (**a**) displacement per atom (dpa) and He concentration in the ZrO_2_–MgO composite irradiated by 50 keV He^+^ ions to a fluence of 1 × 10^16^ ions/cm^2^, and (**b**) displacement per atom, and electronic and nuclear energy loss in the ZrO_2_–MgO composite irradiated by 6.4 MeV Xe^23+^ ions to a fluence of 1 × 10^16^ ions/cm^2^.

**Figure 2 materials-12-02649-f002:**
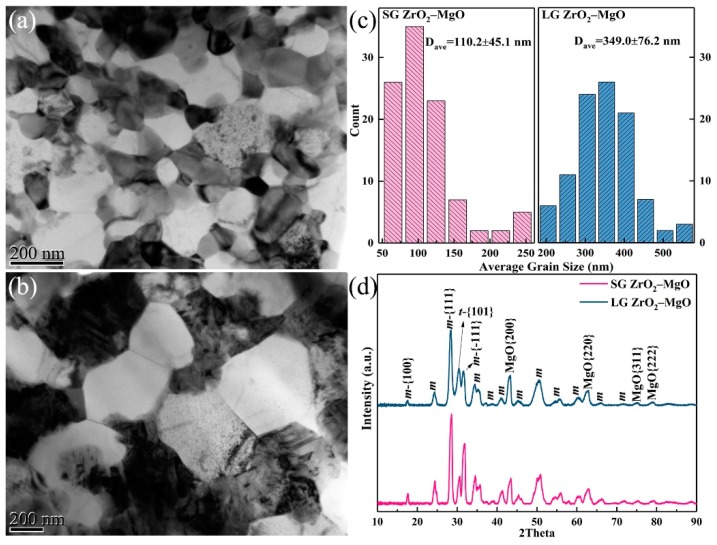
Bright-field transmission electron microscope (TEM) images of the pristine small-grain (SG) (**a**) and large-grain (LG) (**b**) ZrO_2_–MgO composite ceramics sintered by spark plasma sintering (SPS); statistical average grain size of SG and LG ZrO_2_–MgO (**c**), and their crystalline structures measured by grazing incidence X-ray diffraction (GIXRD) (**d**).

**Figure 3 materials-12-02649-f003:**
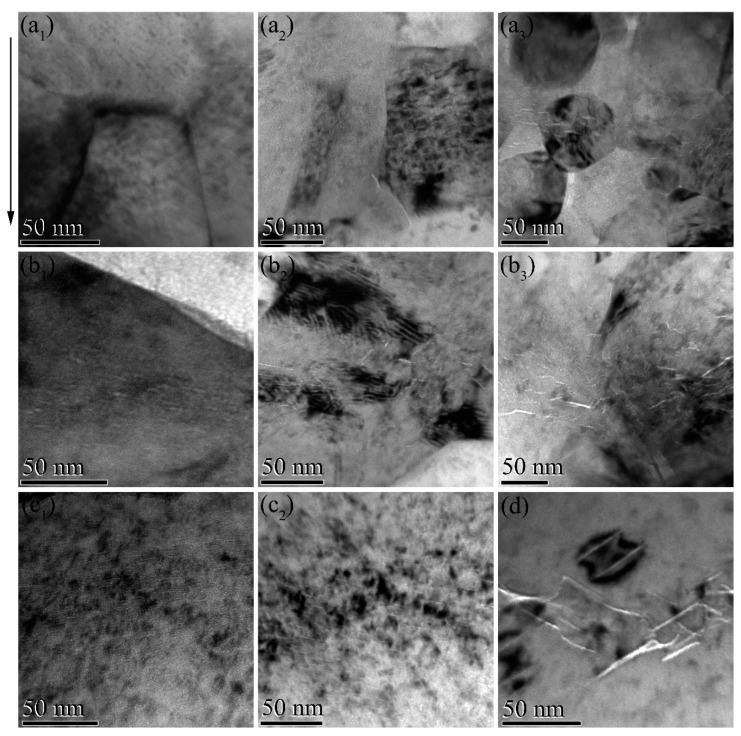
Cross-sectional TEM images of the SGs (**a**), LGs (**b**), MgO (**c**), and yttria-stabilized ZrO_2_ (YSZ) (**d**) irradiated at 773 K by 50 keV He^+^ ions to fluences of 3 × 10^16^ (**a_1_, b_1_, c_1_, d**), 5 × 10^16^ (**a_2_, b_2_, c_2_**), and 1 × 10^17^ (**a_3_, b_3_**) ions/cm^2^, respectively. Black scale bars of different lengths all represent 50 nm. The arrow indicates the direction of ion incidence in the images.

**Figure 4 materials-12-02649-f004:**
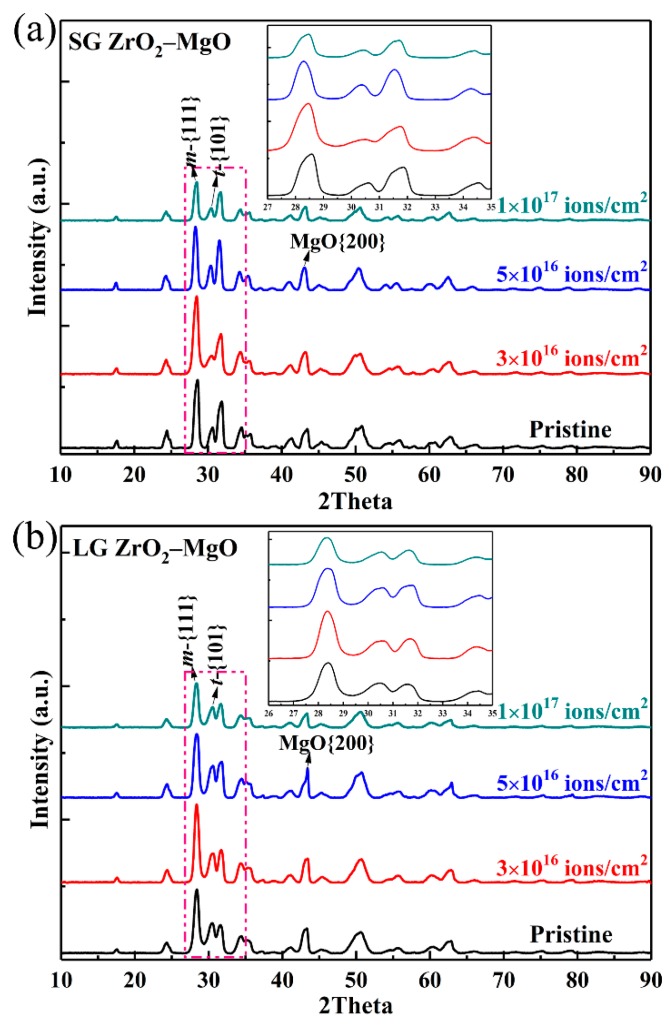
GIXRD patterns of the pristine SGs (**a**) and LGs (**b**) irradiated by 50 keV He^+^ ions to the fluences of 3 × 10^16^, 5 × 10^16^, and 1 × 10^17^ ions/cm^2^ at 773 K; the magnifications of diffraction peaks in the dashed frame are inserted.

**Figure 5 materials-12-02649-f005:**
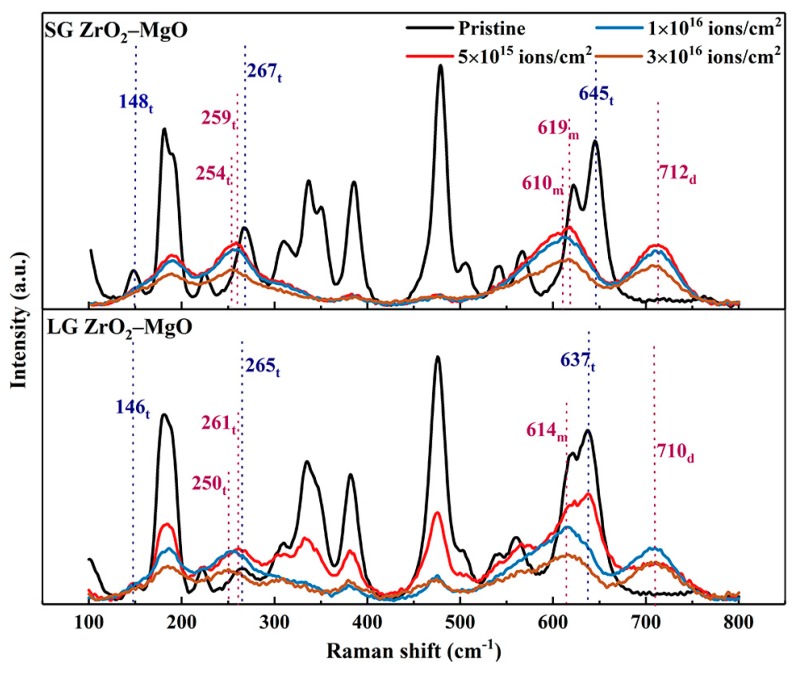
Raman spectra of the pristine SGs and LGs irradiated at RT by 6.4 MeV Xe^23+^ ions to the fluences of 5 × 10^15^, 1 × 10^16^, and 3 × 10^16^ ions/cm^2^. The positions of some bands have been marked, and the lower right letters t, m, and d represent tetragonal phase, monoclinic phase, and defects, respectively.

**Table 1 materials-12-02649-t001:** Diffraction angle 2*θ* (°) values of the {111}*_m_*, {101}*_t_*, and {200}*_MgO_* peaks in the SG and LG samples at different irradiation fluences, obtained from GIXRD patterns. The variation values relative to the pristine peak are listed in parentheses.

Sample	Pristine	3 × 10^16^	5 × 10^16^	1 × 10^17^
SGs—{111}*_m_*	28.42	28.34 (−0.08)	28.28 (−0.14)	28.34 (−0.08)
LGs—{111}*_m_*	28.40	28.36 (−0.04)	28.32 (−0.08)	28.30 (−0.10)
SGs—{101}*_t_*	30.50	30.44 (−0.06)	30.34 (−0.16)	30.40 (−0.10)
LGs—{101}*_t_*	30.40	30.46 (+0.06)	30.46 (+0.06)	30.46 (+0.06)
SGs—{200}*_MgO_*	43.18	43.02 (−0.16)	43.00 (−0.18)	43.04 (−0.14)
LGs—{200}*_MgO_*	43.04	43.16 (+0.12)	43.10 (+0.06)	43.04 (+0)

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
