# Peer review of "Different Radiation Tolerances of Ultrafine-Grained Zirconia–Magnesia Composite Ceramics with Different Grain Sizes"

_materials, 2019, doi:10.3390/ma12172649_

Round 1

Reviewer 1 Report

The manuscript by W. Qin et al. presents results of radiation tolerance study for the case of Zirconia-Magnesia composite ceramics, irradiated with He and Xe ions. The paper is clearly written (except few points in introduction, which will be mentioned later) and represent a solid piece of work where conclusions are justified by experiments. All the performed measurements are well described and explained and I find the results interesting. Therefore, the paper can be accepted for publication after few minor remarks.

1. Please re-read Introduction part and correct few grammar mistakes (some a listed below). The rest of the text does not contain significant mistakes/misprints.

2. Please specify the choice of energy of implanted elements.

3. In the abstract, YSZ is used without full name, it has been introduced only in section 2.1. Please correct.

...methods is reduced its grain size -> ...methods is reducING its grain size;

Nanostructure materials -> NanostrucurED materials;

can effectdively act -> WHICH can effectively act;

By refining... is a widely used -> Refining ... is a widely used;

to amorphous - to amorphize;

the smaller the grain size did not always represent - please correct grammar;

size depended -> size dependent;

nuclear energy filed -> nuclear energy field;

Author Response

The manuscript by W. Qin et al. presents results of radiation tolerance study for the case of Zirconia-Magnesia composite ceramics, irradiated with He and Xe ions. The paper is clearly written (except few points in introduction, which will be mentioned later) and represent a solid piece of work where conclusions are justified by experiments. All the performed measurements are well described and explained and I find the results interesting. Therefore, the paper can be accepted for publication after few minor remarks.

Response: Thank you for these valuable suggestions that have helped us to improve the overall quality of our manuscript. We have carefully considered these comments and made the corresponding correction in our revised manuscript. Please see the “revised manuscript with marked corrections” file.

1. Please re-read Introduction part and correct few grammar mistakes (some a listed below). The rest of the text does not contain significant mistakes/misprints.

...methods is reduced its grain size -> ...methods is reducING its grain size;

Nanostructure materials -> NanostrucurED materials;

can effectdively act -> WHICH can effectively act;

By refining... is a widely used -> Refining ... is a widely used;

to amorphous - to amorphize;

the smaller the grain size did not always represent - please correct grammar;

size depended -> size dependent;

nuclear energy filed -> nuclear energy field;

Response 1: Many thanks for pointing out these errors. We have corrected these errors in the revised manuscript.

2. Please specify the choice of energy of implanted elements.

Response 2: Thanks for your comment. The 50 keV He+ ions irradiation was used to simulate the behaviour of transmutation gas. While the 6.4 MeV Xe23+ ions irradiation was design to simulate the neutron irradiation in fission reactor field.

3. In the abstract, YSZ is used without full name, it has been introduced only in section 2.1. Please correct.

Response 3: Thanks for your suggestion. We have added the full name of YSZ (yttria stabilized ZrO2) in the abstract section. Detailed changes can be seen in revised manuscript.

Reviewer 2 Report

The authors investigated the different radiation tolerance of ultrafine grained zirconia-magnesia composite ceramics with different grain sizes. Two kinds of composite ceramics with different grain sizes were prepared by spark plasma sintering and their radiation damage behaviors were evaluated by helium and xenon ions irradiations. The authors estimated that it is difficult to state which grain size of the composite ceramics has a better radiation tolerance performance under ion irradiation with different energy; therefore, one should consider choosing the suitable grain size depending on the actual application.

I do recommend this manuscript to be accepted for publication in the present form.

Author Response

The authors investigated the different radiation tolerance of ultrafine grained zirconia-magnesia composite ceramics with different grain sizes. Two kinds of composite ceramics with different grain sizes were prepared by spark plasma sintering and their radiation damage behaviors were evaluated by helium and xenon ions irradiations. The authors estimated that it is difficult to state which grain size of the composite ceramics has a better radiation tolerance performance under ion irradiation with different energy; therefore, one should consider choosing the suitable grain size depending on the actual application.

I do recommend this manuscript to be accepted for publication in the present form.

Response: Thank you very much for your affirmation of our work.